REGISTERED REPORT PROTOCOL

# Multivariate prediction of temper outbursts in a sample of youth enriched for irritability using ecological momentary assessment data: A registered report

**Dipta Saha[1], Reut Naim[2,3], Francisco Pereira[4], Melissa A. Brotman[1], Charles Y. Zheng[4]***

**1** Neuroscience and Novel Therapeutics Unit, Emotion and Development Branch, National Institute of Mental Health, National Institutes of Health, Bethesda, Maryland, United States of America, **2** The School of Psychological Sciences, Tel Aviv University, Tel Aviv, Israel, **3** Sagol School of Neuroscience, Tel Aviv University, Tel Aviv, Israel, **4** Machine Learning Core, National Institute of Mental Health, National Institutes of Health, Bethesda, Maryland, United States of America

* charles.zheng@nih.gov

## Abstract

Irritability and temper outbursts are among the most common reasons youth are referred for psychiatric assessment and care. Identifying in vivo clinical variables that precede the onset of temper outbursts would provide valuable clinical utility. Here, we provide the rationale for a study testing the performance of a classifier trained to predict temper outbursts in a group of clinically-referred youth presenting with symptoms of irritability and temper outbursts. Due to the large sample sizes needed for multivariate classification studies, here, we demonstrated the feasibility of our approach using a relatively large preliminary dataset. Our preliminary data included digital based event sampling from an existing Ecological Momentary Assessment dataset consisting of $n = 54$ participants with a total of 932 time points. We used this data to develop a logistic regression-based classifier for predicting the temper outburst prospectively. Our initial evaluation provided encouraging evidence for the possibility of predicting the presence of a temper outburst based on individual's momentary clinical responses (e.g., whether the participant is feeling grouchy, hungry, happy, sad, anxious, tired, etc.) prior to the outburst event, as well as external features (e.g., time of day, day of week). However, due to the risk of false positive discoveries and overfitting, these preliminary results are insufficient to conclusively establish the discovery of predictive rules for irritability in Ecological Momentary Assessment data. To more rigorously assess this classifier, we will collect a large confirmatory set, consisting of at least an additional 20 subjects with an expected total of 400 time points, in which will perform confirmatory analyses of the precision and recall of the classifier already fit using preliminary data. This work will potentially provide the foundation for the identification of features predictive of risk and future development of novel mobile-device-based interventions in youth affected with severe and impairing psychopathology.

**Data availability statement:** We have made available all data for participants who consented for data sharing at the following link: https://osf.io/4ectb/?view_only=22133bc8b94444a2bb84c553de082268. There were 5 participants who did not consent to having their data shared.

**Funding:** This work was supported by the Intramural Research Program (IRP) of the National Institute of Mental Health, National Institutes of Health (NIMH/NIH) ZIA MH002969: Unit on Neuroscience and Novel Therapeutics, and National Institutes of Health (NIMH/NIH) ZIC-MH002968, Machine Learning Core.

**Competing interests:** The authors have declared that no competing interests exist.

# 1. Introduction

Irritability and temper outbursts are one of the most common reasons for child mental health referrals [1–3] and are associated with long-term significant functional impairment and mental health problems [1,4]. Irritability is complex; clinically, it is transdiagnostic and presents across numerous diagnoses [5] and is not a unitary construct but rather consists of discrete dimensions [6]: tonic irritability and phasic irritability. Whereas tonic irritability refers to a baseline cranky, grumpy and negative mood, phasic irritability is associated with discrete behavioral manifestations of anger and temper outbursts. Tonic and phasic irritability are known to have differential relations [7] between and specific mood and anxiety disorders, differences in co-morbidity with ADHD [8] and differing genetic associations [9]. Here, we examine different clinical and behavioral features, measured repeatedly, in vivo, to see if differential clinical symptoms can predict temper outbursts.

The use of real-time, *in vivo* metrics using smartphones and other wireless devices provides the possibility of obtaining naturalistically situated information on the psychological state of children with symptoms of irritability across different clinical conditions (i.e., disruptive mood dysregulation disorder (DMDD), oppositional defiant disorder (ODD), sub-threshold symptoms of DMDD (sub-threshold DMDD), attention deficit hyperactivity disorder (ADHD)) by means of digital-based phenotyping using ecological momentary assessment (EMA) [10–12]. EMA data has the potential to provide the clinician with valuable insight into longitudinally presenting dynamics of irritability and other related clinical symptoms [13]. Our goal in this observational work is to test the degree to which dynamic clinical phenotyping via EMA enables the prospective prediction of temper outbursts in a sample enriched for psychopathology.

Previous work has demonstrated the feasibility of predicting symptomatic behavior from psychological traits [14, 15]. Prior research has demonstrated significant differences in the presentation and manifestation of anger and aggression by sex assigned at birth [9,16]. Extensive prior work has shown significant changes in the trajectory of irritability by age [17, 18]. For example, while temper outbursts are normative during the preschool years, as prefrontal cortex and emotion regulation improves during school age years (i.e., 6-10), normatively temper outbursts decrease [18]. Then, during early adolescence, irritability often increases again during puberty [19]. This increase in anger and irritability during puberty also further interacts with sex assigned at birth [16].

Clinical interactions and prior implementation research have also indicated a pattern of anger and irritability that follows specific temporal relations, including time of day (e.g., post-doc decreased sleep [20], and over the course of the school week [21]. From a clinical perspective, we are aware that negative valence, high arousal emotions (e.g., anxiety and irritability) often co-occur [22] and are distinct from high arousal positively valence emotions (e.g., happiness, excitement). However, it remains unclear the temporal relationship between high arousal positive and negative emotional states [11], as they present, dynamically in vivo and specifically how these other emotions specifically relate to the later presentation of acute temper outbursts.

Therefore, our approach of measuring of irritability at multiple times per day and at multiple time scales could maximize our ability to robustly predict temper outbursts across heterogeneous irritability phenotypes with varying degrees of tonic and phasic irritability, taking into account also the effects of sex, age, and weekly or daily patterns.

Models which succeed at predicting temper outbursts would be valuable not only for potentially revealing predictive relationships between irritability and other time-varying metrics (including other symptoms, time of day, physical states such as hunger), but also because of the possibility to use such models as an early warning system within a device-assisted intervention program to prevent or reduce future outbursts [23]. In this registered protocol, we describe the development of two classification models (regularized logistic regression and

random forest) on existing data, and our planned validation of one of those models (regularized logistic regression) on data that we plan to collect.

**Personalized vs. population-level prediction.** In determining how to formalize the machine learning prediction problem for this project, we considered three different alternatives.

1. Fully personalized prediction, where we train a separate machine learning model for each participant.

2. Personalized prediction with group information, where each participant's model is fit using a combination of their individualized data as well as data from other individuals.

3. Population-level prediction, where all data from all participants is pooled together and a single machine learning model is trained on the pooled dataset.

We ended up deciding on approach 3, population-level prediction, despite the possible advantages of approaches 1 and 2 [24,25]. This is because the more individualized training data is required, the more time families would have to spend on the platform to accumulate training data before they could benefit from interventions. Therefore, we prioritized developing a system with no requirement for individualized training can deliver useful interventions. Furthermore, we have seen in the literature that a population-level approach can indeed produce generalizable predictions from EMA data [26,27] and that, even in cases where personalized prediction is feasible, that population-level approaches can achieve similar performance [25].

## 1.1. Organization of this registered protocol

As the paper adopts a differing organization compared to the typical registered protocol, we now give an overview of the paper's organization. Section 1 is the usual introduction section. The discussion of the preliminary data and analysis for the protocol is given an entire section, Section 2. The protocol for the collection of new data and planned analysis, which is the usual focus of a stage 1 registered report, is provided in Section 3. Section 4 contains the usual discussion section.

The reason for why the preliminary data and its analysis is more extensive than typical is because of our choice to pre-register a trained classifier. By doing so, we can test much more specific hypotheses about the performance of the classifier than would otherwise be possible.

## 1.2. Related work

Naim et al. 2021 [13] describes an experimental design for testing the efficacy of exposure-based cognitive behavior therapy for severe irritability in youth. As part of the experimental design, [13] collects ecological momentary assessment data, both from parental reports and child self-reports. We use the same protocols for recruiting participant families and obtaining child self-reported EMA data as [13]. However, our protocol is observational in nature—we do not collect data from treated patients. Furthermore, our goal is to assess the feasibility of forecasting temper outbursts from EMA data, rather than assessing the effectiveness of any treatment.

## 2. Preliminary analysis

## 2.1. Explanation

### 2.1.1. Goals. The goals of the preliminary data analysis were as follows.

1) Assess the feasibility of training a classifier for predicting a temper outburst from EMA data.

2) Generate hypotheses for the confirmatory study, by means of selecting a promising subset of variables which likely have predictive value for a temper outburst.

3) Develop the actual classifiers that will be assessed in the confirmatory study.

**2.1.2. Data collection procedures.** *Data Collection Procedures.* Recruitment focused on youth aged 8–18 years ($M_{age}$ = 11.6 years, *SD* = 2.2 years; 62.96% male for included participants) meeting criteria for DMDD (*n* = 38), sub-threshold DMDD (*n* = 10), or ODD (*n* = 6). Some of the participants also had co-occurring anxiety disorder(s) and/or attention-deficit hyperactivity disorder (ADHD).

The means of recruitment and exclusion criteria were the same as for the confirmatory sample, as described in section 2.3.2.

This protocol was carried out from July 2018. Participants were recruited from the Washington, DC, metropolitan area in the context of two ongoing studies (ClinicalTrials.gov Identifiers NCT02531893 and NCT00025935). Recruitment strategies leveraged established relationships with local health care providers and schools and included advertising on Facebook and sending postcards to local households. Outpatients were referred to NIMH.

*Sample size rationale.* Using the preliminary dataset, we aimed to develop classifiers that will be tested in the confirmatory sample. As the presence of severe class imbalance increases the sample size needed for classifier training [28], we ensured that we had enough datapoints to contain at least 100 minority class examples.

*Data splitting.* We formed a training set, stage 1a, from the prompts of a subset consisting of 43 out of 54 participants. The validation set, stage 1b, formed from the prompts of a mutually exclusive subset 11 out of 54 participants, was used to assess model performance. The sample size was thus determined by the size of the existing data and the need to split the data to enable unbiased inference of the classifier performance.

## 2.2. Preliminary data

We analyzed a preliminary set of data collected starting July 2018 and ending in December 2022, in which *n* = 57 participants (with a diagnosis of either ODD, DMDD, or sub-threshold DMDD) completed EMA, with a total of 1197 prompts (3 prompts per day for 7 days, up to 21 completed prompts per participant depending on completion rate). Three participants were excluded for having fewer than 3 completed EMA prompts during the recording period (for non-machine learning studies, the inclusion criteria for rate of complete prompts would typically be more strict (e.g., 7 out of 21), but we used a less strict inclusion criteria out of the desire to examine if the prediction could work even on people who fulfilled what we considered as a minimal criterion to evidence engagement by completing 3 out of 21 prompts.)

Data consisting of 1134 prompts from non-excluded participants was included. Out of these prompts, 202 did not have a response, leaving 932 prompts. We completed the collection of preliminary data, inspected the data, and fit and evaluated models to the data. However, due to the relatively low percentage of temper outbursts recorded by participants, we required a relatively large preliminary dataset to ensure that we had sufficient number of temper outbursts to train a classifier (please see detailed information about the sample size in section 2.3.3). Note that the values in Tables Tables 1–3 are based on preliminary data, and hence differ from the statistics in [13].

**2.2.1. Measured variables.**

- *Irritability and related constructs.* As defined in [29], irritability, frustration, aggression and anger are related constructs. The EMA therefore includes four items which are designed to measure these four similar but distinct constructs, thereby allowing a multi-dimensional characterization of irritability. Furthermore, the items are designed to enable irritability chronometry, with three items designed to measure irritability throughout the entire day, and one item assessing irritability at the time of the prompt [13]. Frustration is a normal affective reaction to a blocked goal. Either frustration or threat can give rise to an emotion

**Table 1. Demographic characteristics for participants for preliminary data included in the analysis.**

| Demographic characteristic | |
|---|---|
| **Number of included participants** | $n = 54$ |
| Age range | 8-18 |
| Mean age (SD) | 11.6 (2.2) |
| *Sex assigned at birth (N, %)* | |
| Male | $n = 34$ (63%) |
| Female | $n = 20$ (37%) |
| *Race at birth (N, %)* | |
| Asian | $n = 2$ (4%) |
| Black or African American | $n = 4$ (7%) |
| White | $n = 42$ (78%) |
| Multiple Races | $n = 5$ (9%) |
| Unknown | $n = 1$ (2%) |
| *Ethnicity (N, %)* | |
| Latino or Hispanic | $n = 3$ (6%) |
| Not Latino or Hispanic | $n = 49$ (91%) |
| Unknown | $n = 2$ (4%) |
| *Missing prompts* | |
| Mean (sd) percentage of missing prompts | 17.81% (17.13%) |
| Number (%) of participants with at least one missing prompt | $n = 44$ (81%) |
| *Diagnosis (N, %)* | |
| sub-DMDD | $n = 10$ (19%) |
| DMDD | $n = 38$ (70%) |
| ODD | $n = 6$ (11%) |

**Table 2. Outcome statistics (temper outbursts) for participants for preliminary data included in the analysis.**

| *Temper outbursts* | Overall | Morning | Afternoon | Evening |
|---|---|---|---|---|
| Mean (sd) [min, max] of participant-level frequencies of temper outbursts ($n = 54$) | 0.10 (0.16) [0.0, 0.73] | 0.10 (0.20) [0.0, 1.0] | 0.08 (0.15) [0.0, 0.5] | 0.11 (0.19) [0.0, 0.67] |
| Mean (sd) [min, max] of participant-level frequencies of temper outbursts for male participants ($n = 34$) | 0.06 (0.09) [0.0, 0.33] | 0.05 (0.12) [0.0, 0.57] | 0.05 (0.11) [0.0, 0.5] | 0.07 (0.12) [0.0, 0.57] |
| Mean (sd) [min, max] of participant-level frequencies of temper outbursts for female participants ($n = 20$) | 0.17 (0.21) [0.0, 0.73] | 0.19 (0.27) [0.0, 1.0] | 0.14 (0.18) [0.0, 0.5] | 0.18 (0.24) [0.0, 0.67] |

called anger, which is associated with an increased effort towards goal attainment. Irritability refers to a decreased threshold for experiencing anger in response to frustration. Aggression is a behavioral or verbal expression of anger, which includes temper outbursts.∘ (a) temper outburst – "SINCE the last beep, I felt really, really angry and out of control." (categorical response: "yes" or "no"). *Rationale.* Temper outbursts are acute expressions of anger possibly including physical and verbal aggression which are typically triggered by offensive stimuli.

∘ (b) irritable mood – "SINCE the last beep, aside from being really, really angry and out of control, I was feeling generally grouchy or cranky." (5-point Likert scale, 1 = none of the time; 5 = the whole time). *Rationale.* This item, along with (a) and (c), aims to assess irritability throughout the entire day.

**Table 3. Outcome statistics (other than temper outbursts) for participants for preliminary data included in the analysis.**

| Variable | n (%) for total responses (out of 932 total) | | | | |
|---|---|---|---|---|---|
| | 1 (none of the time) | 2 | 3 | 4 | 5 (the whole time) |
| *Irritability and related constructs (see table 2 for temper outbursts)* | | | | | |
| Irritable mood | 516 (55%) | 212 (23%) | 147 (16%) | 40 (4%) | 17 (2%) |
| Frustration | 482 (52%) | 169 (18%) | 139 (15%) | 80 (9%) | 62 (7%) |
| *Anxiety symptoms* | | | | | |
| Anxious affect | 738 (79%) | 78 (8%) | 71 (8%) | 31 (3%) | 14 (2%) |
| Momentary anxiety | 764 (82%) | 83 (9%) | 47 (5%) | 26 (3%) | 12 (1%) |
| *Momentary feeling state* | | | | | |
| Tired | 309 (33%) | 218 (23%) | 163 (17%) | 126 (13%) | 116 (12%) |
| Happy | 329 (35%) | 163 (17%) | 156 (16%) | 149 (16%) | 135 (14%) |
| Unhappy | 656 (70%) | 116 (12%) | 86 (9%) | 37 (4%) | 37 (4%) |
| Giddy | 681 (73%) | 95 (10%) | 80 (9%) | 48 (5%) | 28 (3%) |
| Hungry | 424 (45%) | 179 (19%) | 164 (18%) | 91 (10%) | 71 (8%) |
| Mood changed | 480 (52%) | 184 (20%) | 152 (16%) | 58 (6%) | 58 (6%) |

○ (c) frustration – "SINCE the last beep, I felt frustrated." (5-point Likert scale, 1 = not at all; 5 = extremely).*Rationale.* This item, along with (a) and (b), aims to assess irritability throughout the entire day.

○ (d) momentary anger – "At the time of the beep, I felt ANNOYED or ANGRY." (5-point Likert scale, 1=not at all; 5= extremely). *Rationale.* This item measures the emotion of anger in the current moment. Note: due to high correlation with item (a) temper outburst, this item was not used in the prediction models.

• Anxiety symptoms. These measure aspects of anxiety as operationalized by the Pediatric Anxiety Rating Scale [30].

○ (a) anxious affect – "SINCE the last beep, I felt worried or scared." (5-point Likert scale, 1 = not at all; 5 = extremely). *Rationale.* This item measures the overall level of anxiety experienced over the last few hours (in between ratings).

○ (b) anxious avoidance – "SINCE the last beep, I avoided doing things because I felt worried or scared." (categorical yes or no). *Rationale.* This item measures the prevalence of avoidance behaviors caused by anxiety.

○ (c) momentary anxiety [31] – "At the time of the beep, I felt WORRIED or SCARED." (5-point Likert scale, 1 = not at all; 5 = extremely). *Rationale.* This item measures the momentary level of anxiety.

• The participant's momentary feeling state. These measures were collected to assess emotional lability [11], which may be associated with depression, anxiety, and attention-deficit/hyperactivity disorder. Since the goal of these measures is to assess change in the presence of the feeling, these measures all elicit the momentary feeling state, except for item f, which directly asks for the participant's self-assessment of emotional lability.

○ (a) tired -- "At the time of the beep, I felt TIRED." (5-point Likert scale, 1 = not at all; 5 = extremely). *Rationale:* We included this item to investigate potential associations between

physical factors like being tired to the likelihood of experiencing elevated irritability or outbursts.

◦ (b) happy -- "At the time of the beep, I felt HAPPY." *Rationale:* We included this item to be able to measure the degree of positive emotion lability (by observing changes in this variable.)

◦ (c) unhappy -- "At the time of the beep, I felt unhappy, sad, or miserable." *Rationale:* We included this item to be able to measure the degree of negative emotion lability (by observing changes in this variable.)

◦ (d) giddy -- "At the time of the beep, I felt much more giddy, silly, or happy than usual." *Rationale:* We included this item to be able to measure the degree of positive emotion lability (by observing changes in this variable.)

◦ (e) hungry -- "At the time of the beep, I felt HUNGRY." *Rationale:* We included this item to investigate potential associations between physical factors like being hungry to the likelihood of experiencing elevated irritability or outbursts.

◦ (f) having experienced a change in mood -- "SINCE the last beep, my mood changed a lot." (5-point Likert scale, 1 = not at all; 5 = extremely). *Rationale*: This item measures emotional lability experienced throughout the day.

## 2.3. Methods for classifier development on preliminary data

**Timing Considerations.** Since one of our goals is to test the feasibility of developing an early warning and intervention system based on EMA data, we will structure the classifier to act on data that would be available to predict an outburst before it occurs. The EMA data is collected three times a day, and during each collection period, all the questions are collected in the same short session. This includes both questions that ask for the state of the respondent (e.g., anxious, frustrated, etc.) since the time of the last beep, as well as questions that ask for the respondent's momentary state (e.g., momentary anger, hunger, etc.). Both may be predictive of an impending outburst.

**Variables used for prediction.** All the models considered predict the temper outburst from time-lagged variables provided as their input. We explored two different lag settings, which we call "lag-1" and "lag-2".

The "lag-1" model uses the variables from the previous prompt to predict the response to the temper outburst question from the current prompt. For example, to predict the temper outburst response on day 4 afternoon, we use the responses from day 4 morning as inputs to the prediction model. To predict the temper outburst response on day 3 morning, we use the responses from day 2 evening. If day 2 evening is completely missing, then we use the next most recent prompt, e.g., day 2 afternoon.

The "lag-2" model incorporates all the variables used by the "lag-1" model and adds to it all the variables from the prompt preceding the previous prompt. For example, to predict the temper outburst response on day 3 morning, we use the responses from day 2 evening and day 2 afternoon. Meanwhile, the "lag-3" model adds the variables from three prompts ago in addition to all the variables from the "lag-2" model. Continuing with our example, to predict the temper outburst response on day 3 morning, we use the responses from day 2 morning, in addition to the responses from day 2 afternoon (also used by "lag-2") and day 2 evening (also used by "lag-2" and "lag-1").

The variables coincident with the endorsement of a temper outburst (lag = 0) are not used in the prediction because, given the design of the EMA application, temper outbursts will typically only be recorded after they have already occurred. Hence, such measurements would not be available to an early warning intervention system.

**Logistic Regression.** We used logistic regression with balanced class weights, and evaluated both L1 and L2 regularization in the process of choosing a regularizer (see below for details on hyperparameter selection). Furthermore, we used Bayesian comparison of classification models [32] to perform backwards variable selection. Both hyperparameter and variable selection were carried out within the training set.

To select hyperparameters, we used stratified cross-validation with 40 splits to search over a hyperparameter grid. The number of splits was chosen to achieve a balance between having a large number of splits (which improves classifier performance) and ensuring that each split had a minimal number of temper outbursts (e.g., 1 or 2). We picked the hyperparameter combination (penalty + cost parameter) that minimized the F2 score. The F2 score was used to give more weight to recall than to precision, as we consider it acceptable to have a high percentage of false alarms in order to achieve a high coverage of prospective temper outbursts.

We used the following grid parameters:

- "C" (cost parameter, inverse of alpha parameter): {.001,.01,.1,1,10,100,1000}

- penalty: L1, L2

We used scikit-learn [33] to implement the regression.

**Random Forest.** On the same set of variables, we also developed a balanced random forest classifier. As with logistic regression, we used balanced class weights, and used stratified 40-fold cross-validation to pick hyperparameters. The hyperparameters were:

- Number of trees: {5,10,20,50,100,120}

- Maximum depth: {1,2,3,4,5,6,7,8}

- Split criterion: Gini or Entropy

We used Imbalanced-learn [34] to implement the random forest.

**Feature selection.** We used Bayesian model comparison [32] to perform feature selection for the logistic regression classifier using the training set (the data collected in the first stage). Specifically, we used the Bayesian paired t-test with one-sided posterior probabilities. We did not use the random forest classifier to perform feature selection, but rather used the feature set obtained from logistic regression for both classifier families.

In order to carry out backwards feature selection, we used the following procedure.

Given a model (list of features), we:

1) compared (using the Bayesian paired t-test) the model to all versions of the model with one variable deleted,

2) for any model with a deleted variable having a higher than 50% probability of being better than the original model, we took the model with the highest probability of being better than the original model, and repeated step 1 with that model.

We stopped this procedure when none of the deleted features had at least a 50% posterior probability of performing better than the current model. The resulting model was therefore selected as the final classifier to be evaluated on the confirmatory sample.

## 2.4. Results of classifier development on preliminary data

**2.4.1. Variable selection.** We considered 23 variables and eliminated six of them by means of a backwards selection algorithm using Bayesian comparison tests [32], as described in section 2.3. The following 17 variables were selected.

-- The participant's age and self-reported sex assigned at birth.

-- The participant's momentary state of feeling: tired, worried, happy, grouchy, angry, frustrated, worried, unhappy, having experienced a change in mood.

-- Whether the day of the week is Sunday, Monday, Tuesday, Wednesday, Saturday, or Sunday

-- Whether the time of day is evening.

We excluded the following variables from the model based on the results of variable selection:

-- Whether the participant has been hungry, or felt giddy.

-- Day of week: Thursday and Friday.

-- Time of day: morning and afternoon.

**2.4.2. Logistic regression.** We determined the optimal number of time lags (1, 2, or 3) to use for logistic regression. The AUC for lag-1 was 0.73, the AUC of lag-2 was 0.78, and the AUC of lag-3 was 0.77. Hence, we selected lag-2 for regularized logistic regression. Detailed results of this analysis are included in S2 Appendix.

The best regularization scheme for logistic regression, as determined using 40-fold cross-validation within the training set, was L2 regularization with C = 0.01. In table 4, we present the model performance for regularized logistic regression, applied to the test examples.

Precision, also called positive predictive value, is defined as the fraction of positive predictions that were true positives (the fraction of times where a temper outburst was endorsed out of the times that the model predicted a temper outburst). Recall, also called sensitivity, is defined as the fraction of positive examples that were predicted to be positive (the fraction of times where the model predicted a temper outburst out of the times in which a temper outburst was endorsed). F1-score is the harmonic mean of precision and recall, given by the formula F1 = 2 * (precision * recall)/ (precision + recall). Support is the total number of test set observations in each category.

Table 5 presents the model weights for logistic regression, as well as Bayesian 95% credible intervals for the variables. We use the Pymc package [35] to implement the Bayesian inference, with N(0,400) prior for the intercept and N(0,0.01) prior (with sigma^2 = 0.01 chosen to match the selected hyperparameter C = 0.01) for each of the non-intercept beta coefficients. For MCMC, we used the NUTS sampler [36] with 10,000 draws and an acceptance target of 0.99.

A positive (negative) value indicates that a YES response is associated with increased (decreased) chance of temper outburst in the next measurement period. Note that the disagreement between the regularized logistic regression intercept and Bayesian interval for the intercept are due to the use of the class balanced cost function for regularized logistic regression. For questions corresponding to labels, please see Table 5.

**Table 4. Prediction performance of temper outbursts for DMDD, sub-threshold DMDD, and ODD patients in held-out test examples of existing dataset, using regularized logistic regression.**

|  | Precision | Recall | F1-score | Support |
|---|---|---|---|---|
| No outburst | 0.97 | 0.82 | 0.89 | 154 |
| Temper outburst | 0.25 | 0.69 | 0.37 | 13 |

**2.4.3. Random forest.** The best parameters for random forest, as determined by 40-fold cross-validation, were to have an ensemble of 20 estimators, maximum depth of 2, and entropy as the split criterion. The resulting classifier had an AUC of 0.70, which we considered insufficient to warrant continued development of the random forest classifier. Hence, we only plan to evaluate the logistic regression classifier in the planned confirmatory study.

## 2.5. Preliminary data availability

We have made available all data for participants who consented to data sharing at the following link: https://osf.io/4ectb/?view_only=22133bc8b94444a2bb84c553de082268. There were 5 participants who did not consent to having their data shared.

# 3. Confirmatory data collection and analysis protocol

## 3.1. Hypotheses

In an EMA digital platform that queries real-time symptom metrics three times a day [13] and in a model that is shared across participants, we hypothesize that it is possible to prospectively predict a temper outburst in the following assessment based on the following measurements:

-- The participant's age and self-reported sex assigned at birth.

-- The participant's momentary state of feeling: tired, worried, happy, angry, frustrated, having been worried, unhappy, having experienced a change in mood.

-- Whether the day of the week is Monday, Tuesday, Wednesday, Thursday/Friday, Saturday, or Sunday. *Remark*: the reason why Thursday and Friday are considered as one category is because those two variables were eliminated as an outcome of feature selection on preliminary data (see section 2.4.1). Hence, any observation which does not have the indicator 1 for any of the remaining days (Monday, Tuesday, Wednesday, Saturday, Sunday) can be considered in the "left-out" group which consists of all observations occurring on Thursday or Friday.

**Table 5. Model weights and intercept for L2-regularized logistic regression, and 95% credible intervals from Bayesian regression.**

| Variable (lag) | Coefficient | Bayesian 95% credible interval | Variable (lag) | Coefficient | Bayesian 95% credible interval |
|---|---|---|---|---|---|
| happy (1) | -0.20 | [-0.26, 0.12] | happy (2) | -0.08 | [-0.21, 0.18] |
| unhappy (1) | 0.10 | [-0.11, 0.27] | unhappy (2) | 0.12 | [-0.11, 0.27] |
| grouchy (1) | 0.13 | [-0.12, 0.27] | grouchy (2) | 0.07 | [-0.13, 0.25] |
| frustrated (1) | 0.08 | [-0.12, 0.27] | frustrated (2) | 0.03 | [-0.14, 0.25] |
| angry (1) | 0.25 | [-0.04, 0.34] | angry (2) | 0.22 | [-0.05, 0.32] |
| mood_changed (1) | 0.04 | [-0.15, 0.24] | mood_changed (2) | -0.01 | [-0.16, 0.21] |
| worried_now (1) | 0.11 | [-0.12, 0.27] | worried_now (2) | 0.11 | [-0.11, 0.28] |
| been_worried (1) | 0.08 | [-0.13, 0.25] | been_worried (2) | 0.09 | [-0.12, 0.27] |
| tired (1) | 0.05 | [-0.14, 0.24] | tired (2) | 0.07 | [-0.12, 0.26] |
| age | -0.08 | [-0.20, 0.18] | Monday | -0.11 | [-0.23, 0.15] |
| female | 0.17 | [-0.05, 0.32] | Tuesday | -0.10 | [-0.22, 0.16] |
| Night | -0.01 | [-0.16, 0.21] | Wednesday | -0.01 | [-0.19, 0.19] |
| *(intercept)* | -0.16 | [-2.76, -2.03] | Saturday | -0.06 | [-0.21, 0.17] |
|  |  |  | Sunday | 0.02 | [-0.17, 0.20] |

-- Whether the time of day is evening, or before evening (morning and afternoon being considered as one category). *Remark*: the reason why morning and afternoon are considered as one category is because those two variables were eliminated as an outcome of feature selection on preliminary data (See above Remark).

### 3.2. Design plan

**3.2.1. Study design.** *Study Type.* The study is an observational study, since data is collected from study participants that are not randomly assigned to a treatment.

*Blinding.* No blinding is involved in this study. The authors had access to information that could identify participants during and after collection of data.

*Study design.* The preliminary data were collected according to a protocol from [13]. No participants were treated during the data collection. Quoting from [13]:

"Youth participants enriched for symptoms of irritability and temper outbursts and an identified parent completed a clinical evaluation visit before being enrolled in the research protocol. Participants subsequently completed a standardized EMA training session during which a research assistant familiarized the participant with the smartphone and protocol and reviewed each EMA item by guiding the participant through a practice prompt. To enhance feasibility and compliance, for each day during the upcoming 7 days of EMA, participants preselected 60-min periods during standardized time windows within which prompts would be delivered: morning/before school (6:00–9:00 a.m.), afternoon/after school (3:00–6:00 p.m.), and evening/before bedtime (7:00–10:00 p.m.). The actual prompt times were randomized within these time periods... Participants used either a personal or a study- provided smartphone...

Following the training session, participants were prompted three times per day for seven consecutive days. At each prompt, participants received a text message with a link to the website through which the items were delivered. Once the prompt was received, participants had 60 min to complete the assessment before it expired and was considered incomplete."

**3.2.2. Data collection procedures.** Recruitment will focus on youth aged 8–18 years meeting criteria for DMDD, sub-threshold DMDD, or ODD.

The means of planned recruitment and exclusion criteria will be identical to the procedures described in [13], as quoted:

"Participants were recruited via direct mailings and online advertisements. Participants were evaluated for eligibility and diagnostic status by a doctoral- or master's-level clinician using the Schedule for Affective Disorders and Schizophrenia for School-Age Children – Present and Lifetime version [37]. Primary diagnosis was based on the chief presenting complaint and clinician judgment of the most severely impairing diagnosis. Recruitment focused on youth whose irritability was chronic and not clearly related to another ongoing or episodic diagnosis (e.g., major depressive disorder, bipolar disorder). Exclusion criteria were: IQ < 70, assessed using the Wechsler Abbreviated Intelligence Scale [38]; a diagnosis of posttraumatic stress disorder, schizophrenia, neurological disorder, developmental disorder, bipolar disorder, or obsessive–compulsive disorder; a current major depressive episode; or substance abuse within 3 months of participation. Participants and their parents provided written assent and consent, respectively. Participants were compensated and offered a monetary bonus for completing ≥75% of prompts. The study was approved by the National Institute of Mental Health Institutional Review Board."

A second wave of data collection, for the confirmatory set, is currently ongoing at NIMH.

**3.2.3. Sample size.** In the confirmatory data set (to be completed by approximately 2027), we plan to collect 20 participants, with an expected total of 300.79 prompts. We determined this prospective sample size based on a power analysis with simulated data.

We simulated data based on a simulation ground truth logistic regression model with L2 penalization, and penalization constant tuned to produce data that would result in a cross-validated accuracy of around 0.40, matching what we saw in the real data. Then, by resampling without replacement participants from the training set, and drawing labels according to the predictions of the simulation ground truth model, we produced synthetic datasets with sample sizes ranging from 1 to 34 participants. For each target sample size, we drew 1000 synthetic data sets of that size, and we assessed the power in terms of the average proportion of coefficients with incorrectly estimated signs using regularized balanced logistic regression. Detailed results are presented in S1 Appendix.

Based on the results of the power analysis, there was a consensus that, when taking into account both the clinical and methodological considerations, the more conservative would be 25 participants, even though a case could be made to justify 15 participants. Our consensus decision is to take the average of these two extremes, and therefore our new plan is to stop collecting participants for this analysis once we have collected 20 participants, which results in around 300.79 prompts on average and has a 31% average sign error for estimating coefficients (Table in S1 Appendix). For this number of prompts, assuming the same proportion of prompts will contain outbursts as in the exploratory sample (10%), we expect to see 30.08 temper outbursts on average.

### 3.3. Statistical models

**3.3.1. Data.** The full list of selected variables is as follows
Response variable: Temper outburst (binary)
Predictor variables:

1. non-time-varying:
    1. Demographic variables: age and self-reported sex assigned at birth.

2. time-varying:
    1. Indicator variables from previous time point: giddy, tired, worried, happy, grouchy, angry, frustrated, worried, unhappy, change in mood.
    2. Time of day and day of week. See section 2.2.1, "Measured Variables" for details.

While we used data-splitting for model development in the analysis of the preliminary data, in the confirmatory dataset we will use the entire confirmatory dataset as a test set to evaluate the logistic regression classifier we have already developed.

**Transformations.** The day of week will be extracted from the timestamp of the raw data.

Recall that the data will be collected for 7 days, 3 times a day per observation period, with some participants completing multiple observation periods (details in sampling plan). The time of day will be obtained from the column number of the raw data (as columns 1-21 are the 21 consecutive EMA prompt periods, with morning, afternoon, and evening for each of 7 days). The participant's age will be rounded to integer values.

All predictors will be rescaled to lie between 0 and 1 using min-max scaling. Observations with missing entries will be excluded from the analysis. Since we do not use observations with partial entries, we will report the missing cases on an observation-by-observation basis rather than a variable-by-variable basis.

**3.3.2. Analytic approach.** We will evaluate one pre-fitted model, L2-regularized logistic regression, as developed in section 3.

We will use the "lag-2" model as described in section 2.3. Hence, time-varying predictors will be included with both lags of 1 and 2, so that information in the two measurement periods proceeding a temper outburst is used to predict the absence/presence of a temper outburst. All variables will be scaled with min-max scaling to lie between 0 and 1.

**3.3.3. Inference criteria.** We will compute posterior credible intervals for the precision and recall of the model [39]. Specifically, we will obtain the following statistics:

1. TP (number of true positives) = True temper outburst that is predicted correctly.

2. FP (number of false positives) = Lack of temper outburst that is predicted as outburst.

3. TN (number of true negatives) = Lack of temper outburst that is predicted correctly.

4. FN (number of false negatives) = True temper outburst that is predicted as lack of outburst.

This allows us to perform Bayesian inference on the precision and recall as described in [32]. Using a prior lambda = 0.5 – the uninformative Jeffreys prior -- the posterior distributions of the precision and recall are as follows.

$$Precision \sim Beta\big(TP + 0.5. FP + 0.5\big)$$

$$Recall \sim Beta\big(TP + 0.5, FN + 0.5\big)$$

The posterior credible intervals can be obtained using the quantiles of these posterior distributions. Let QBeta($q$, $a$, $b$) denote the quantile function for the Beta distribution with quantile $q$ and shape parameters $a$, $b$. Hence, the 95% posterior credible interval for precision is

$$Precision: \big[QBeta\big(0.025, TP + 0.5. FP + 0.5\big), QBeta\big(0.975, TP + 0.5. FP + 0.5\big)\big]$$

$$Recall: \big[QBeta\big(0.025, TP + 0.5. FN + 0.5\big), QBeta\big(0.975, TP + 0.5. FN + 0.5\big)\big]$$

*Expected results.* Based on our preliminary results, where the regularized logistic regression model achieved a precision of 0.25 and recall of 0.69 for temper outbursts, we expect that the 95% posterior credible interval for the precision will contain a left endpoint greater than 0.1 and a right endpoint less than 0.4. We expect that the 95% posterior credible interval for recall will contain a left endpoint greater than 0.55 and a right endpoint less than 0.95.

**3.3.4. Exploratory analyses.** We include two main categories of exploratory analysis in this protocol. First is a post-hoc analysis of the performance of individualized prediction models. Second is an optional set of diagnostic analyses to be performed only if the result of the confirmatory analysis deviates from the preliminary analysis by a pre-defined margin.

**Individualized prediction.** Individualized prediction involves taking the model trained on all participant data in the training set and then "fine-tuning" the model by optimizing for a few iterations on the data from an individual participant in the test set. Technically, this is accomplished by using the "warm-start" argument in the scikit-learn logistic regression class, combined with a low number of maximum iterations. The model individualized this way will then be compared to the original model in terms of performance on unseen data from that participant. This reflects a use case where an existing model would be adapted to a new participant using data collected at the beginning of a trial, and then used to generate predictions from that point onwards.

**Diagnostic analyses.** There are several outcomes of the main analysis that would warrant further diagnostic analyses of the data. We are including these potential analyses in this section, since it is customary for a pre-registration to include planned exploratory analyses.

These are the scenarios which would trigger our diagnostic analyses.

1) If the AUC of the classifier on the confirmatory dataset differs from the AUC of the classifier on the preliminary dataset by more than 0.05.

2) If the upper bound of the posterior credible interval for any of the following variables is less than the nominal value on the preliminary dataset. The variables are: precision for outburst, precision for non-outburst, recall for outburst, recall for non-outburst.

3) If the width of the posterior credible interval for any of the variables listed in (2) above exceeds 0.5.

The diagnostic analyses include:

i)   Checking for distributional shift, by comparing the means and pairwise correlations of all variables between the preliminary dataset vs the test set.

ii)  Checking for unusual participants, by modeling the joint distribution of the variables between participants as a mixture-of-multinomials model and looking for participants with low likelihood in the mixture-of-multinomials.

iii) Checking if any of the variables included in the model are particularly to blame for poor generalization, by fitting models on the training set with one variable removed at a time and seeing if removing any of the variables this way allows for improved performance on the test set.

We believe the results of these analyses would be important in explaining any deviation from our expected results, and helpful to other researchers interested in carrying out similar experiments.

## 4. Discussion

In this registered protocol, we propose a study for assessing the potential of machine-learning-based methods for enabling multivariate prediction of temper outbursts using ecological momentary assessment data.

Identifying the extent to which temper outbursts can be predicted could be employed in future interventions for pediatric irritability, such as informing pediatric patient, parents and treating clinicians to the child's clinical status. In future work, EMA reports could also be paired with real-time monitoring of youth's physiology and behavior to derive a more complete phenotype of irritability and develop precise temporally-informed intervention targets.

One limitation to our study is that all time periods (morning, afternoon, night) are treated equivalently by our modeling approach. However, in practice the delay between night and morning is the longest and, while the delay between afternoon and night is usually longer than the delay between morning and afternoon, there is some variability as participants are free to choose when to complete the prompt within the allowed window. Therefore, it is plausible that there could be differences between the three times of day in terms of the optimal prediction rule. However, when we attempted to fit time-of-day-specific prediction models, we saw no statistically significant difference in the performance of time-of-day specific models and the model that we developed based on all times of day (Table A in S2 Appendix). Yet, given more data it might be possible to develop time-of-day-specific classifiers for differing times of day that would outperform a time-of-day-general classifier.

There are two additional major limitations to this proposed study. Firstly, the sample will be nonrepresentative, as the study consists of individuals who choose to participate in NIMH studies. To mitigate the bias due to the nonrepresentative study population, we will recruit across the community. Furthermore, some biases may be due to participation being potentially limited by factors that facilitate remote research compliance. Reasons for lack of participation include finding the task to be boring or annoying, or conflict between the task and existing activities. To increase compliance, we will allow participants to adjust the scheduling of the responses within 3-hour intervals for morning, noon, and evening, and randomized the timing of the prompts within the selected 60-minute interval for each time period. However, we recognize that the freedom to select the response period may lead to additional bias due to possible confounding between preferences for timing windows and measured behavior.

## Supporting information

**S1 Appendix. Power analysis.**
(DOCX)

**S2 Appendix. Results on differing time lags.**
(DOCX)

## Acknowledgments

We are grateful to Simone Haller and Arshitha Basavaraj for assistance with data sharing, and Lauren Henry for proofreading. The authors thank the participants and their families for their time and commitment to help us carry out the research in this study

## Author contributions

**Conceptualization:** Dipta Saha, Melissa A. Brotman, Francisco Pereira, Charles Y. Zheng.

**Data curation:** Reut Naim.

**Formal analysis:** Dipta Saha, Charles Y. Zheng.

**Funding acquisition:** Melissa A. Brotman.

**Investigation:** Dipta Saha.

**Methodology:** Dipta Saha, Francisco Pereira.

**Project administration:** Melissa A. Brotman.

**Resources:** Reut Naim.

**Software:** Dipta Saha.

**Supervision:** Melissa A. Brotman, Francisco Pereira, Charles Y. Zheng.

**Validation:** Reut Naim.

**Visualization:** Dipta Saha.

**Writing – original draft:** Charles Y. Zheng.

**Writing – review & editing:** Reut Naim, Melissa A. Brotman, Charles Y. Zheng.

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
