## [Decision Letter · Decision Letter 0]

14 Dec 2023

PONE-D-23-20323Multivariate prediction of temper outbursts in youth enriched for irritability using Ecological Momentary Assessment dataPLOS ONE

Dear Dr. Zheng,

Thank you for submitting your manuscript to PLOS ONE. After careful consideration, we feel that it has merit but does not fully meet PLOS ONE’s publication criteria as it currently stands. Therefore, we invite you to submit a revised version of the manuscript that addresses the points raised during the review process.

**ACADEMIC EDITOR: ** There was some difficulty in securing reviewers for this submission. On a careful reading of the comments of the reviewer, I believe many of them can be addressed through a major revision. In particular, the reviewer's comments on the design choices need to be addressed in depth in my view.

The reviewer has mentioned their lack of expertise in machine learning. However in my reading of the report, the authors only intend to use a logistic regression model for further analysis. I have two comments in this regard, the first relates to the the choice of performance metrics for the logistic regression and the random forest classifiers. The AUC of the random forest is mentioned to be .54 and is the reason given for not using it for further analysis. No AUC is given for the logistic regression for comparison. The second relates to Table 3. While the weights for each variable is given, its also normal to provide p-values corresponding to each variable, to demonstrate which variables are significant at a particular alpha level.

We look forward to receiving your revised manuscript.

Kind regards,

Sandip Varkey George, PhD

Academic Editor

PLOS ONE

Journal Requirements:

Upon re-submitting your revised manuscript, please upload your study’s minimal underlying data set as either Supporting Information files or to a stable, public repository and include the relevant URLs, DOIs, or accession numbers within your revised cover letter. For a list of acceptable repositories, please see http://journals.plos.org/plosone/s/data-availability#loc-recommended-repositories . Any potentially identifying patient information must be fully anonymized.

Important: If there are ethical or legal restrictions to sharing your data publicly, please explain these restrictions in detail. Please see our guidelines for more information on what we consider unacceptable restrictions to publicly sharing data: http://journals.plos.org/plosone/s/data-availability#loc-unacceptable-data-access-restrictions .  Note that it is not acceptable for the authors to be the sole named individuals responsible for ensuring data access.

3. We note that the original protocol that you have uploaded as a Supporting Information file contains an institutional logo. As this logo is likely copyrighted, we ask that you please remove it from this file and upload an updated version upon resubmission.

Reviewers' comments:

Reviewer's Responses to Questions

**Comments to the Author**

1. Does the manuscript provide a valid rationale for the proposed study, with clearly identified and justified research questions?

Reviewer #1: Partly

2. Is the protocol technically sound and planned in a manner that will lead to a meaningful outcome and allow testing the stated hypotheses?

Reviewer #1: Partly

3. Is the methodology feasible and described in sufficient detail to allow the work to be replicable?

Reviewer #1: Yes

4. Have the authors described where all data underlying the findings will be made available when the study is complete?

Reviewer #1: Yes

5. Is the manuscript presented in an intelligible fashion and written in standard English?

Reviewer #1: Yes

6. Review Comments to the Author

You may also provide optional suggestions and comments to authors that they might find helpful in planning their study.

Reviewer #1: This is a review of a registered report protocol that proposes to use machine-learning on EMA-assessments (and age and sex) to extract features that predict next-moment temper outbursts in diagnosed youths. The protocol has merit, with a preliminary data analysis to support the choices for the data collection and analysis for the proposed study.

However, I have several comments – major and minor – regarding the rationale and description of the methods in this protocol. Briefly summarized, I was happy to see a high degree of detail in their model descriptions, but I judged that the rationale and literature embedding of this manuscript needed further work. Some major design choices and their implications/limitations also deserved more detailed explication. Note that I am not an expert on machine-learning techniques, and thus my review pertains mainly to the conceptual and EMA-design aspects of the protocol. My detailed comments are listed below.

Major:

First, regarding the outlined rationale for the study. I believe that gaining a better understanding of the contextual conditions that contribute to temper outbursts in youths is a valid and worthwhile research goal. However, from the current Introduction it is unclear to me what is already known on this topic from clinical practice and theory in this area. Are there any existing (treatment) models that have been described that this research directly feeds into or expands? Without such context it is hard for me to judge to what extent this research design is a worthwhile way to investigate this topic.

Another question I have is regarding the chosen time lag. In the Introduction, the authors describe that EMA data may provide “insight into longitudinally-presenting dynamics of irritability and other related clinical symptoms”. However, the proposed analyses only adopt a lag-1 design (section 5.1.2.). Lag-2 and lag-3 might still contain highly relevant predictive information. Have the authors considered examining other lags?

Relatedly, the prediction from the last beep of the day (e.g., day 1 beep 3) to the first beep of the next day (day 2 beep 1; overnight) might be qualitatively different from the association between beeps on the same day. If I understand correctly, these overnight lags are left unaltered. Have the authors considered how the inclusion of overnight lags might affect their model results? I consider it pertinent for the authors to make their considerations explicit in the text, as well as mention this in the limitations.

Moreover, unequal spacing between the other within-day beeps may also affect the predictive ability of closer-spaced beeps (e.g., afternoon to evening is likely closer than morning to afternoon). Please add some discussion on this.

Another point that I think requires further discussion is the fact that the proposed study aims to find group-level predictors, for within-person processes. In the Introduction, the authors already speculate about the potential for clinical applications and early warning systems through apps. However, this all rests on the assumption that their model can find generic factors from across a small group of individuals that will apply well enough to individual cases. I would like to see a deeper embedding of this in the introduction and discussion. Could the authors describe why they think it is likely that they would find generalizable results with their method?

Some comments on section ‘2.3.2. Sample size’. I would like to see a more detailed explanation of the target sample size. Based on the description it seems that approximately 40 outbursts occurring in the span of 400 measurements is considered sufficient to run the confirmatory analyses. Can the authors provide further explanation and references to support that target number? I am satisfied that the 10% expectation comes from their own preliminary data but would like to see more justification for the chosen number of observations.

Also, there is no mention of the within-person and temporal dependency that these observations have, which will limit the predictive contribution of a given observation. Since some children will have no outbursts, and some may have many, this influences how much information they can draw from each person. And, since the model aims to predict next-moment temper outbursts, the beep prior to an event is almost as important as the beep with the outburst itself. Please make note of this in the manuscript and how it has affected the design choices.

In section 4.1. the authors refer readers to their previous manuscript for the rationale behind the variables. I appreciate that paraphrasing methods sections from previous work can be tedious – especially when direct replication is the goal – and I understand that a lot of overlap exists between their earlier work and the current protocol. However, a direct explanation of the rationale for the chosen variables seems important enough. Please briefly summarize the chosen conceptualization for your measures.

Minor:

In the abstract:

- The authors describe the sample as “a group of clinically-referred youth enriched for symptoms of irritability and temper outbursts”. I do not understand what the word ‘enriched’ is supposed to mean in this context, perhaps it is a jargon term. Please consider clarifying this language.

- The authors describe the total number of EMA time points in their preliminary sample, and intended data collection. Since the expected number of temper outbursts is crucial to the study, the authors could consider making note of that too.

§ 2.1

- The authors include a parenthetical in their hypotheses to say “Thursday and Friday being considered as one category.” I have found no clear explanation for this anywhere in the manuscript. If this is based on their prior work, they could simply add that here. If it is due to methodological limitations, please clarify that.

§ 2.3.1. Explanation of existing data

- Could the authors make explicit what they considered ‘not enough data’ for inclusion?

- In Table 1, the mean number of complete ratings are reported. Please also report the range and SD.

§ 2.3.2.

- The aims for the data collection are listed here, with the age and gender distribution listed at 3 decimal precision. I found this rather confusing initially, especially because the values differ from those reported in the quoted paper and Table 1 reports only age at one decimal precision. Please make explicit here that these numbers are based on the values found in the preliminary data, and consider rounding them up.

- “The means of planned recruitment and exclusion criteria will be similar to those…” Are they similar or identical. If similar, please be explicit in what will be changed, have any steps been updated since the prior data collection?

- Note that the next section on Sample size is also labeled 2.3.2., please correct.

§ 3.5

- I found the color scale of Figure 1 to be a little hard to read, given that it lacks a clear differentiation around 0. The figure note describes a ‘strong negative correlation’ for some of the variables, but I found the dark purple of 0 and near-black of -0.2 to not be very differentiable, and found this negative end of the scale additionally tricky to judge given much wider range of positive correlations. Consider replotting this with a diverging color scale (e.g., see here: https://r-graph-gallery.com/38-rcolorbrewers-palettes.html)

§ 4.1 Measured variables

- Please explain the chosen use of capital letters in Table 3 and the description of the variables in section 4.1. Were the prompts presented literally with those words capitalized, or were they perhaps presented in bold? Some further description is advised.

7. PLOS authors have the option to publish the peer review history of their article (what does this mean? ). If published, this will include your full peer review and any attached files.

**Do you want your identity to be public for this peer review?** For information about this choice, including consent withdrawal, please see our Privacy Policy .

Reviewer #1: No

---

## [Author Response · Author response to Decision Letter 1]

19 Apr 2024

We are thankful to the academic editor and the reviewer for their insightful comments. In addressing these comments, we were invited to think more deeply about the motivation of this work and the particular choices we made, and to perform additional analyses to justify our decisions. In some cases, we even discovered new findings that spurred substantial changes to our paper. We think these changes greatly improve the paper, and so we are profoundly grateful to the editor and reviewer for stimulating these developments. We also hope that our responses and our additions to the paper will address the important concerns raised but also improve the readability and utility of the paper for readers.

Response to academic editor

> "The AUC of the random forest is mentioned to be .54 and is the reason given for not using it for further analysis. No AUC is given for the logistic regression for comparison. "

Thank you for pointing this out. We found out that we could include an additional 8 training participants (not new data--these were 8 out of the 54 previously described participants) and this increased the AUC of random forest to 0.70. The AUC of the lag-1 logistic regression increased to 0.73 with the additional participants and we report this in the S2 appendix. Please note however that we have switched to a lag-2 logistic regression model in this revision, which has an AUC of 0.78.

> ."...Table 3. While the weights for each variable is given, its also normal to provide p-values corresponding to each variable, to demonstrate which variables are significant at a particular alpha level."

Thank you for bringing this to our attention, as we generally agree that reporting of p-values or other forms of uncertainty quantification are essential for the reporting of statistical modeling results. For the Bayesian model selection we used, there is no existing method to provide frequentist p-values. However, it is possible to provide Bayesian posterior credible intervals for the parameters, which have similar properties to frequentist confidence intervals (Albers et al. 2018). In section 3.4.2, we describe this approach. We included these credible intervals in the updated Table 3.

Response to reviewer 1

> "...from the current Introduction it is unclear to me what is already known on this topic from clinical practice and theory in this area. Are there any existing (treatment) models that have been described that this research directly feeds into or expands? Without such context it is hard for me to judge to what extent this research design is a worthwhile way to investigate this topic."

We agree with the importance of indicating existing research. We have expanded the list of cited references which we discuss in the introduction.

> "...he proposed analyses only adopt a lag-1 design (section 5.1.2.). Lag-2 and lag-3 might still contain highly relevant predictive information. Have the authors considered examining other lags?"

Thank you for this excellent suggestion! In response to it, we fit the regularized logistic regression model with lags 1, 2, and 3 as described in section 3.4.2. Based on these results, we have revised the manuscript to pre-register the lag-2 logistic regression model instead of the lag-1 logistic regression model.

> Relatedly, the prediction from the last beep of the day (e.g., day 1 beep 3) to the first beep of the next day (day 2 beep 1; overnight) might be qualitatively different from the association between beeps on the same day. If I understand correctly, these overnight lags are left unaltered. Have the authors considered how the inclusion of overnight lags might affect their model results? I consider it pertinent for the authors to make their considerations explicit in the text, as well as mention this in the limitations.

Moreover, unequal spacing between the other within-day beeps may also affect the predictive ability of closer-spaced beeps (e.g., afternoon to evening is likely closer than morning to afternoon). Please add some discussion on this.

Thank you for these excellent questions on the number of lags to include in the model, as well as the effects of different times of day. We conducted additional experiments on these questions, and include results and discussion in the S2 appendix. As a result of these investigations, we decided to switch from lag-1 to lag-2 regularized logistic regression, which we describe in section 3.4.2.

We have included the following text in section 6.

"One limitation to our study is that all time periods (morning, afternoon, night) are treated equivalently by our modeling approach. However, in practice the delay between night and morning is the longest, and while the delay between afternoon and night is usually longer than the delay between morning and afternoon, there is some variability as participants are free to choose when to complete the prompt within the allowed window. Therefore, it is plausible that there could be differences between the three times of day in terms of the optimal prediction rule. However, when we attempted to fit time-of-day-specific prediction models, we saw no statistically significant difference in the performance of time-of-day specific models and the model that we developed based on all times of day (see S2 Appendix). Yet, given more data it might be possible to develop time-of-day-specific classifiers for differing times of day that would outperform a time-of-day-general classifier."

> Another point that I think requires further discussion is the fact that the proposed study aims to find group-level predictors, for within-person processes. In the Introduction, the authors already speculate about the potential for clinical applications and early warning systems through apps. However, this all rests on the assumption that their model can find generic factors from across a small group of individuals that will apply well enough to individual cases. I would like to see a deeper embedding of this in the introduction and discussion. Could the authors describe why they think it is likely that they would find generalizable results with their method?

You are right, it is important to explain why we decided to predict only using population data and not customizing for individuals. In summary, only a population-level approach allows immediate generalizability to a new user without requiring a period of initial data collection. We discuss this in greater detail within a discussion of individualized versus population-level prediction approaches in section 3.3.

> Some comments on section ‘2.3.2. Sample size’. I would like to see a more detailed explanation of the target sample size. Based on the description it seems that approximately 40 outbursts occurring in the span of 400 measurements is considered sufficient to run the confirmatory analyses. Can the authors provide further explanation and references to support that target number? I am satisfied that the 10% expectation comes from their own preliminary data but would like to see more justification for the chosen number of observations.

Thank you for raising the important issue of sample size justification. Originally, our estimate of 400 completed EMA ratings was based on intuition gained from working with the preliminary data. However, in response to the reviewer we conducted a more rigorous power analysis to check if our intuition was well-calibrated. We describe this power analysis in section 2.3.3.

> Also, there is no mention of the within-person and temporal dependency that these observations have, which will limit the predictive contribution of a given observation. Since some children will have no outbursts, and some may have many, this influences how much information they can draw from each person. And, since the model aims to predict next-moment temper outbursts, the beep prior to an event is almost as important as the beep with the outburst itself. Please make note of this in the manuscript and how it has affected the design choices.

You have raised an insightful question on the choice of whether to use variables concurrent with the predicted temper outburst. We have clarified our analysis choices and how they are tailored to the goal of developing an intervention system in section 5.1.2, where we comment that "[t]he variables coincident with a temper outburst are not used in the prediction, because given the design of the EMA application, temper outbursts will typically only be recorded after they have already occurred. Hence, such measurements would not be available to an early warning intervention system."

> In section 4.1. the authors refer readers to their previous manuscript for the rationale behind the variables. I appreciate that paraphrasing methods sections from previous work can be tedious – especially when direct replication is the goal – and I understand that a lot of overlap exists between their earlier work and the current protocol. However, a direct explanation of the rationale for the chosen variables seems important enough. Please briefly summarize the chosen conceptualization for your measures.

You are right, it is important for us to explain the conceptualization of our measures. Our prompt design is based on the differentiation of irritability into tonic and phasic irritability, which we explain in the introduction.

> - The authors describe the sample as “a group of clinically-referred youth enriched for symptoms of irritability and temper outbursts”. I do not understand what the word ‘enriched’ is supposed to mean in this context, perhaps it is a jargon term. Please consider clarifying this language.

Thanks for noting our use of jargon, and consequently a way to improve the readability of our paper for a more general audience. We have rewritten that sentence of the abstract to read, "Here, we provide the rationale for a study to test the performance of a classifier trained to predict temper outbursts in a group of clinically-referred youth presenting with symptoms of irritability and temper outbursts."

> - The authors describe the total number of EMA time points in their preliminary sample, and intended data collection. Since the expected number of temper outbursts is crucial to the study, the authors could consider making note of that too.

Thank you for bringing this to our attention. We now note the number of expected outbursts in our intended data collection.

>§ 2.1

- The authors include a parenthetical in their hypotheses to say “Thursday and Friday being considered as one category.” I have found no clear explanation for this anywhere in the manuscript. If this is based on their prior work, they could simply add that here. If it is due to methodological limitations, please clarify that.

Indeed, we did not include any explanation for this methodological decision, and it is important for us to explain how this was not an arbitrary choice but in fact was a consequence of the feature selection procedure we used (described in section 3.3). In fact, the reason why Thursday and Friday end up being considered as one category is because our feature selection procedure eliminated the dummy variables for Thursday and Friday. We now include this explanation in section 2.1.

> § 2.3.1. Explanation of existing data

- Could the authors make explicit what they considered ‘not enough data’ for inclusion?

Thank you for this excellent question! We included the following text in section 2.3.1. "Three were excluded for having fewer than 3 completed EMA ratings during the recording period. (For non-machine learning studies, the inclusion criteria for rate of complete ratings would typically be more strict (e.g 7/21), but we used a less strict inclusion criteria out of the desire to examine if the prediction could work even on people with low compliance. We considered 3 ratings out of 21 to be a minimal criterion to evidence engagement.)"

> - In Table 1, the mean number of complete ratings are reported. Please also report the range and SD.

Thank you for this valuable suggestion! In the process of computing the standard deviations, we noticed that we reported incorrect numbers for the number of prompts used. We now report the corrected average number of complete prompts and the standard deviation in Table 1. Incidentally, we now consistently use the word "prompt" to refer to the data collected from a single participant from one out of three daily sessions ather than "rating".

> § 2.3.2.

- The aims for the data collection are listed here, with the age and gender distribution listed at 3 decimal precision. I found this rather confusing initially, especially because the values differ from those reported in the quoted paper and Table 1 reports only age at one decimal precision. Please make explicit here that these numbers are based on the values found in the preliminary data, and consider rounding them up.

Thank you for bringing this formatting issue to our attention. We have decided to use 3 decimal precision (whenever available) throughout the paper for the sake of consistency. Also, we have included the following text in section 2.3.2.

"Note that the values in Table 1 are based on preliminary data, and hence differ from the statistics in Naim et al. 2021."

> - “The means of planned recruitment and exclusion criteria will be similar to those…” Are they similar or identical. If similar, please be explicit in what will be changed, have any steps been updated since the prior data collection?

Good question. They are identical, and this is now indicated in section 2.3.2.

> - Note that the next section on Sample size is also labeled 2.3.2., please correct.

Thank you for bringing this to our attention. We have corrected the section numbering.

>§ 3.5

- I found the color scale of Figure 1 to be a little hard to read, given that it lacks a clear differentiation around 0. The figure note describes a ‘strong negative correlation’ for some of the variables, but I found the dark purple of 0 and near-black of -0.2 to not be very differentiable, and found this negative end of the scale additionally tricky to judge given much wider range of positive correlations. Consider replotting this with a diverging color scale (e.g., see here: https://r-graph-gallery.com/38-rcolorbrewers-palettes.html)

Thank you for this suggestion! We have changed the color scale of Figure 1 to a diverging color scale, and indeed it seems much easier to interpret compared to the previous version of the figure.

> § 4.1 Measured variables

- Please explain the chosen use of capital letters in Table 3 and the description of the variables in section 4.1. Were the prompts presented literally with those words capitalized, or were they perhaps presented in bold? Some further description is advised.

Thank you for bringing this lack of explanation to our attention. Yes, the prompts were presented literally with those words capitalized and we have noted this in the table caption.

We will greatly appreciate feedback on whether we have addressed most of the concerns raised by the academic editor and the reviewer, and we look forward to any additional questions, suggestions or comments.

---

## [Decision Letter · Decision Letter 1]

1 Aug 2024

PONE-D-23-20323R1Multivariate prediction of temper outbursts in youth enriched for irritability using Ecological Momentary Assessment dataPLOS ONE

Dear Dr. Zheng,

Thank you for submitting your manuscript to PLOS ONE. After careful consideration, we feel that it has merit but does not fully meet PLOS ONE’s publication criteria as it currently stands. Therefore, we invite you to submit a revised version of the manuscript that addresses the points raised during the review process.

We look forward to receiving your revised manuscript.

Kind regards,

Sandip Varkey George, PhD

Academic Editor

PLOS ONE

Additional Editor Comments:

The reviewers, particularly reviewer 3, have given extensive comments relating to the structuring of the manuscript. Despite changes made during the previous round of revision reviewer 3 still feels the manuscript could benefit from substantial restructuring and clarifications. In addition, please also address the point raised by reviewer 2 on how the present manuscript relates to reference [15].

Reviewers' comments:

Reviewer's Responses to Questions

**Comments to the Author**

1. Does the manuscript provide a valid rationale for the proposed study, with clearly identified and justified research questions?

Reviewer #2: Yes

Reviewer #3: Yes

2. Is the protocol technically sound and planned in a manner that will lead to a meaningful outcome and allow testing the stated hypotheses?

Reviewer #2: Partly

Reviewer #3: Partly

3. Is the methodology feasible and described in sufficient detail to allow the work to be replicable?

Reviewer #2: Yes

Reviewer #3: Yes

4. Have the authors described where all data underlying the findings will be made available when the study is complete?

Reviewer #2: Yes

Reviewer #3: Yes

5. Is the manuscript presented in an intelligible fashion and written in standard English?

Reviewer #2: No

Reviewer #3: Yes

6. Review Comments to the Author

You may also provide optional suggestions and comments to authors that they might find helpful in planning their study.

Reviewer #2: • Please include the study method (observational, protocol) in the title

• I am confused about how this is a protocol when the study has one already, cited as [15], or whether this is a preliminary investigation. The tense jumps between past and present and the method section describes both a past study and a future study. If this is a protocol study, then it might make more sense to present the new protocol for the ongoing/confirmatory study (assuming that it is different from 15) and the planned analyses, and then include the preliminary analyses as a result. I am also unclear as to whether the new findings (i.e., Our initial evaluation provided encouraging evidence for the possibility of predicting the presence of a temper outburst based on individual’s momentary clinical responses…) is suitable for a protocol.

• Please avoid long paragraphs of quotes as per the method.

• The rationale for why day of week and time of day should relate to anger belongs in the introduction

• I am unclear as to why no other exploratory analyses will be conducted with the new dataset?

Reviewer #3: Comments to authors:

This is an interesting approach with potential translation into clinical practice at long term that employs EMA data to prospectively predict the occurrence of temper outbursts in children and adolescents with (supposedly) elevated levels of irritability. I have revised a revised version of the original submission. My comments are based on the revised version and the responses given to the editor and the referee. Overall, I found that this work needs to be much clearer, transparent, and streamlined. Currently, it is hard to read due to the way it is structured, since the first sections refer to relevant information that is not presented until later in the manuscript. That the text pivots back and forth between the preliminary data and the confirmatory data make is extremely confusing. As asked by the previous referee, but not addressed, I think it would be good referring less to the previous paper, and provide more information in the current one to make it more understandable and a stand alone piece. Please see my comments below:

1. First, the manuscript needs restructuring to be understandable. My suggestion is moving the whole point 3 (Classified development using preliminary data) before point 2. In addition, the current point, 2.3.1 (Explanation of existing data) should be merged under current point 3 (which, as I said, I think it should be point 2). Otherwise, talking about confirmatory data before describing the preliminary data makes it so hard to read. Moreover, you later talk about the confirmatory data again, repeating information in some instances. It could be simplified all together. I understand authors are following the OSF Pregistrered reports template, but presenting all related to the preliminary data before presenting anything else would make a huge difference. Mostly because all the hypotheses, selection of predictors, analyses etc… are a result of your preliminary data.

2. In the abstract, it is said that the preliminary data consisted on 57 participants with 1197 time points, but this is not accurate. The authors describe how 3 participants were excluded because they had less than 3 prompts (out of 21) completed. Later in the methods the authors mention that “Data consisting of 1134 prompts from non-excluded participants was included”, this suggests that they had 0% of missing. Is that correct? It is hard to believe that 3 participants had less than 3 prompts and the other participants had all 21 completed. Please report % of missing prompts.

3. In the current manuscript, it is not until second Table 3 that the reader finds out how the key variables used in this study were collected. My suggestion is, when moving section 3 to be section 2, under data collection, describe data collection procedure using EMA (as you have it under 2.3 Design plan> study design) and then describe thoroughly how each of the variables was measured, as in the second Table 3, and as in 4.1 Measured variables. This explanation of the variables should go way earlier before you explain any results or models, or analyses. Also, the authors should be explicit in that there are some variables that are momentary, but many others are collected retrospectively asking “SINCE the last beep…. This is important. In the models, which variables are used to predict a Temper oubtursts reported since the last prompt? The, for example, change in mood since the last prompt, or the change in mood since the last prompt from the previous prompt? Be clear about this.

4. There are two Tables 3. I would remove the second table. If variables are well described well in advance as asked, that table is not necessary.

5. The referee says “In section 4.1. the authors refer readers to their previous manuscript for the rationale behind the variables. I appreciate that paraphrasing methods sections from previous work can be tedious – especially when direct replication is the goal – and I understand that a lot of overlap exists between their earlier work and the current protocol. However, a direct explanation of the rationale for the chosen variables seems important enough. Please briefly summarize the chosen conceptualization for your measures.” However, this comment is not addressed. Yes, the authors make the differentiation between phasic and tonic irritability in the introduction, but still there is no rationale for WHY these variables and not others were selected in the first place to be tested for potential predictors of temper outbursts. What is the evidence for each of these variables? What led the authors to include time of day, or day of the week as a potential predictor? And sex? And age? And happiness? And anxiety? And so on… This should be included either in the introduction or at least, in the methods when describing the variables earlier in the manuscript.

6. Related to point 3 and 5, it is said that “The variables coincident with a temper outburst are not used in the prediction, because given the design of the EMA application, temper outbursts will typically only be recorded after they have already occurred. Hence, such measurements would not be available to an early warning intervention system.” Depending how predictors are included in the model, this statement would also apply to all variables collected with “SINCE the last beep..” including irritable mood, frustration, anxious affect, anxious avoidance, grouchy and change in mood.

7. The authors keep referring to paper [15] but I think they should be referring to paper [11] instead. Am I right? Or [15] is correct?

8. Table 1 is odd. I would expect this table to be the first part of the results of current point 3.4. This table should be intended to provide descriptive statistics of the participants, indicating Mean and SD for continuous variables, and number of observations and percentages n(%) for categorical variables. There is no heading. Include race/ethnicity information, and other sociodemographic variables if available. We need to know what type of participants the manuscript is analyzing. Include diagnoses too. Ideally, the table should include descriptive statistics for all variables employed in the models. The most important are temper outbursts, but the information provided for these is unclear (Mean and SD of frequency?). What is the denominator of such frequency? I would be transparent; report Mean and SD of temper outbursts, and then mean and SD of frequency (I believe is the number of temper outbursts/number of prompts completed). Also report the range across participants.

9. The previous information is important to interpret the results of the current paper and for the following; It is interesting to see the seemingly low number of temper outbursts. For example, the authors could not estimate robustly the effects of time of day because low number of outbursts per time of day. I wonder whether there were enough number of temper oubursts per day of the week to properly test the effect of these. This is the kind of information that should be reported on Table 1. Most importantly, most participants meet criteria for DMDD being one of the criterion presenting 3 or more temper outbursts per week. I wonder whether this criterion is met. It is unclear because, as metioned before, information on temper oubtursts is not transmitted fully.

10. As suggested by the referee, 3 decimal reporting is not customary. APA guidelines suggest 1 decimal for Means and SD of integer scales, 2 decimal for other statistics, 2 or 3 decimal for exact p-value unless p<.001.

11. In the current point 2.3.2 Data collection procedure, there is confusion on the reporting of ages. The referee also mentions this section, because it clearly leads to confusion. The sentence is written in future tens, because describes future data collection. How is therefore possible to report the Mean age and SD and % of males of the future participants? These numbers belong to the preliminary data only.

12. Why the confirmatory dataset is expected to collect approximately 300 data points from 20 participants if these are collected using 3 prompts over 7 days? Wouldn’t be this close to 420 data points? Or this is estimation based on the % of missing data points in the preliminary sample that the authors do not report?

13. The rationale for prioritizing a population-level prediction approach should be provided way in advance in the introduction. All this text “we prioritized developing a system with no requirement for individualized training can deliver useful interventions. Furthermore, we have seen in the literature that a population-level approach can indeed produce generalizable predictions from EMA data [21, 22] and that even in cases where personalized prediction is feasible, that population-level approaches can achieve similar performance [20].” Should be provided in the background, instead of nearly at the end of the manuscript.

14. Under sample size rationale of current point 3.2 it is said that “our preliminary data had 12 participants with 1 week of data, and 39 participants with 2 weeks of data.”. This is unclear; I thought that data was collected over 7 days only, but here 2 weeks are mentioned.

15. The current point 3.5 Correlational analysis of preliminary data, which provides Figure 1 with the colored matrix, it is unclear why this analysis, which is not declared when describing statistical approaches, is carried out. The section says that “During the model development process, we examined the pairwise correlations between variables in order to identify pairs of variables that were likely to be redundant. We found that "happy" and "unhappy" were highly negatively correlated, and that "unhappy", "grouchy" and "frustrated" were highly positively correlated, as we show in Fig 1.” But that’s it. So what is the conclusion? Are these redundant or not? If they are, what are the authors going to do about it? Authors should declare the analyses, justify it, and then interpret the results.

7. PLOS authors have the option to publish the peer review history of their article (what does this mean? ). If published, this will include your full peer review and any attached files.

**Do you want your identity to be public for this peer review?** For information about this choice, including consent withdrawal, please see our Privacy Policy .

Reviewer #2: No

Reviewer #3: No

---

## [Author Response · Author response to Decision Letter 2]

28 Oct 2024

Response letter

To the academic editor and reviewers:

We are thankful to the academic editor and the reviewer for their insightful comments. In response, we have re-organized the structure of the paper, greatly improving its readability.

Response to academic editor

> The reviewers, particularly reviewer 3, have given extensive comments relating to the structuring of the manuscript. Despite changes made during the previous round of revision reviewer 3 still feels the manuscript could benefit from substantial restructuring and clarifications. In addition, please also address the point raised by reviewer 2 on how the present manuscript relates to reference [15].

Indeed, these comments on changing the structure of the paper were very helpful, and we have re-organized the paper to be organized into separate sections for the preliminary data already collected and the confirmatory data that is planned to be collected. We also clarify the relation of the manuscript to Naim et al. 2021, as we elaborate below in our response to the reviewer #2.

Response to reviewer #2

> Please include the study method (observational, protocol) in the title

Thank you for the suggestion. We have edited the title to "Multivariate prediction of temper outbursts in an observational study of youth enriched for irritability using Ecological Momentary Assessment data."

> I am confused about how this is a protocol when the study has one already, cited as [15], or whether this is a preliminary investigation.

Please note that we incorrectly cited [15] for the data collection protocol. We have revised the paper to reflect the correct citation, [11]. We have added a new section on pages 4-5, "1.2. Related Work" in which we clarify the relation of this study to [11]. This is an important point that we neglected to make fully clear in the previous version.

"Naim et al. 2021 [11] describes an experimental design for testing the efficacy of exposure-based cognitive behavior therapy for severe irritability in youth. As part of the experimental design, [11] ecological momentary assessment data was collected, both from parents/caregiver and children. We use the same protocols for recruiting participant families and obtaining child EMA data as [11]. However, our protocol is observational in nature -- we do not collect data from treated patients. Furthermore, our goal is to assess the feasibility of forecasting temper outbursts from EMA data, rather than assessing the effectiveness of any treatment."

> The tense jumps between past and present and the method section describes both a past study and a future study. If this is a protocol study, then it might make more sense to present the new protocol for the ongoing/confirmatory study (assuming that it is different from 15) and the planned analyses, and then include the preliminary analyses as a result. I am also unclear as to whether the new findings (i.e., Our initial evaluation provided encouraging evidence for the possibility of predicting the presence of a temper outburst based on individual’s momentary clinical responses…) is suitable for a protocol.

We are extremely grateful for the suggestion by reviewers #2 and #3 to re-organize our paper by clearly separating the preliminary data analysis and the protocol for confirmatory data (please see pages 6-20 for preliminary data analysis and pages 20-29 for protocol). We have re-structured our paper and think that it should read much more clearly now.

• Please avoid long paragraphs of quotes as per the method.

We are not sure why the reviewer is opposed to long quotes. There is no copyright issue, since the quoted text is from a published paper which we cite and believe it help the reader by giving a more context and relevant information.

• The rationale for why day of week and time of day should relate to anger belongs in the introduction

Thank you for this suggestion. As we elaborate further in our response to reviewer #3, question 5, we have added our rationale for including these measures in the introduction.

• I am unclear as to why no other exploratory analyses will be conducted with the new dataset?

Thank you for this question. Our initial purpose of the confirmatory dataset is primarily to validate the predictive performance of the classifier developed using the preliminary dataset. However, there is a hypothetical scenario where the performance on the new dataset was drastically different than the performance on the preliminary dataset. Hence, we have now included these potential post-hoc diagnostic analyses and post-hoc analyses of inter-individual variability in the new section 3.3.4., Exploratory Analyses of the revised manuscript, on pages 27-29.

Response to reviewer #2

> This is an interesting approach with potential translation into clinical practice at long term that employs EMA data to prospectively predict the occurrence of temper outbursts in children and adolescents with (supposedly) elevated levels of irritability. I have revised a revised version of the original submission. My comments are based on the revised version and the responses given to the editor and the referee. Overall, I found that this work needs to be much clearer, transparent, and streamlined. Currently, it is hard to read due to the way it is structured, since the first sections refer to relevant information that is not presented until later in the manuscript. That the text pivots back and forth between the preliminary data and the confirmatory data make is extremely confusing. As asked by the previous referee, but not addressed, I think it would be good referring less to the previous paper, and provide more information in the current one to make it more understandable and a stand alone piece. Please see my comments below:

> 1. First, the manuscript needs restructuring to be understandable. My suggestion is moving the whole point 3 (Classified development using preliminary data) before point 2. In addition, the current point, 2.3.1 (Explanation of existing data) should be merged under current point 3 (which, as I said, I think it should be point 2). Otherwise, talking about confirmatory data before describing the preliminary data makes it so hard to read. Moreover, you later talk about the confirmatory data again, repeating information in some instances. It could be simplified all together. I understand authors are following the OSF Pregistrered reports template, but presenting all related to the preliminary data before presenting anything else would make a huge difference. Mostly because all the hypotheses, selection of predictors, analyses etc… are a result of your preliminary data.

We are extremely grateful for the suggestion by reviewers #2 and #3 to re-organize our paper by clearly separating the preliminary data analysis and the protocol for confirmatory data. We have re-structured our paper and think that it should read much more clearly now.

> 2. In the abstract, it is said that the preliminary data consisted on 57 participants with 1197 time points, but this is not accurate. The authors describe how 3 participants were excluded because they had less than 3 prompts (out of 21) completed. Later in the methods the authors mention that “Data consisting of 1134 prompts from non-excluded participants was included”, this suggests that they had 0% of missing. Is that correct? It is hard to believe that 3 participants had less than 3 prompts and the other participants had all 21 completed. Please report % of missing prompts.

Thank you for this suggestion! You are correct, in the abstract we include the excluded participants. We have now revised it to reflect only the included participants. As for the remaining 1134 prompts, there were 202 which were completely missing, leaving 932 prompts. We now report the number of non-missing prompts for included participants in the abstract and the percentage of missing prompts in table 1a on pages 9-10.

> 3. In the current manuscript, it is not until second Table 3 that the reader finds out how the key variables used in this study were collected. My suggestion is, when moving section 3 to be section 2, under data collection, describe data collection procedure using EMA (as you have it under 2.3 Design plan> study design) and then describe thoroughly how each of the variables was measured, as in the second Table 3, and as in 4.1 Measured variables. This explanation of the variables should go way earlier before you explain any results or models, or analyses. Also, the authors should be explicit in that there are some variables that are momentary, but many others are collected retrospectively asking “SINCE the last beep…. This is important. In the models, which variables are used to predict a Temper oubtursts reported since the last prompt? The, for example, change in mood since the last prompt, or the change in mood since the last prompt from the previous prompt? Be clear about this.

These are great suggestions! We agree with the idea of describing the details of the collected variables in proximity of the description of the data collection protocol. Hence, we now have moved the previous section 4.1 to section 2.2.1, on pages 10-15. Since the previous table 3 is redundant with the description in 2.2.3, we have deleted the table. We have also now clarified the timing of the variables used to predict a temper outburst in section 2.3. Please see our response to your question 6 for this point.

> 4. There are two Tables 3. I would remove the second table. If variables are well described well in advance as asked, that table is not necessary.

Agreed -- we have removed the second table.

> 5. The referee says “In section 4.1. the authors refer readers to their previous manuscript for the rationale behind the variables. I appreciate that paraphrasing methods sections from previous work can be tedious – especially when direct replication is the goal – and I understand that a lot of overlap exists between their earlier work and the current protocol. However, a direct explanation of the rationale for the chosen variables seems important enough.

Thank you for this suggestion. We appreciate that adding the rationale in the current paper is appropriate and improves the readability. We have added those details in section 2.2.1 on pages 10-15, which was previously section 4.1.

> Please briefly summarize the chosen conceptualization for your measures.” However, this comment is not addressed. Yes, the authors make the differentiation between phasic and tonic irritability in the introduction, but still there is no rationale for WHY these variables and not others were selected in the first place to be tested for potential predictors of temper outbursts. What is the evidence for each of these variables? What led the authors to include time of day, or day of the week as a potential predictor? And sex? And age? And happiness? And anxiety? And so on… This should be included either in the introduction or at least, in the methods when describing the variables earlier in the manuscript.

These are excellent questions, and we have revised the introduction to discuss the background research underpinning our selection of these measures. Specifically, on pages 3-4, we now state:

"Previous work has demonstrated the feasibility of predicting symptomatic behavior from psychological traits [12, 13]. Prior research has demonstrated significant differences in the presentation and manifestation of anger and aggression by sex assigned at birth [33, 34]. Extensive prior work has shown significant changes in the trajectory of irritability by age [36, 37]. For example, while temper outbursts are normative during the preschool years, as prefrontal cortex and emotion regulation improves during school age years (i.e., 6-10), normatively temper outbursts decrease [37]. Then, during early adolescence, irritability often increases again during puberty [35]. This increase in anger and irritability during puberty also further interacts with sex assigned at birth [34].

Clinical interactions and prior implementation research have also indicated a pattern of anger and irritability that follows specific temporal relations, including time of day (e.g., post-doc decreased sleep [31], and over the course of the school week [30]. From a clinical perspective, we are aware that negative valence, high arousal emotions (e.g., anxiety and irritability) often co-occur [38] and are distinct from high arousal positively valence emotions (e.g., happiness, excitement). However, it remains unclear the temporal relationship between high arousal positive and negative emotional states [9], as they present, dynamically in vivo and specifically how these other emotions specifically relate to the later presentation of acute temper outbursts."

Furthermore, we have included rationale for each of our individual measures on pages 10-15 in section 2.2.1., "Measured variables".

> 6. Related to point 3 and 5, it is said that “The variables coincident with a temper outburst are not used in the prediction, because given the design of the EMA application, temper outbursts will typically only be recorded after they have already occurred. Hence, such measurements would not be available to an early warning intervention system.” Depending how predictors are included in the model, this statement would also apply to all variables collected with “SINCE the last beep..” including irritable mood, frustration, anxious affect, anxious avoidance, grouchy and change in mood.

We appreciate the reviewer's careful attention to the timing of the variables collected, and we see that the paper could benefit from increased clarity on the timing of the data collection and when prediction could potentially occur within an intervention system. We have therefore added a paragraph in section 2.3. that addresses the timing of the variables used for the outburst prediction. Specifically, on pages 15-16, we state:

"Since one of our goals is to test the feasibility of developing an early warning and intervention system based on EMA data, we will structure the classifier to act on data that would be available to predict an outburst before it occurs. The EMA data is collected three times a day, and during each collection period, all the questions are collected in the same short session. This includes both questions that ask for the state of the respondent (e.g. anxious, frustrated, etc.) since the time of the last beep, as well as questions that ask for the respondent's momentary state (e.g. momentary anger, hunger, etc.). Both may be predictive of an impending outburst. "

> 7. The authors keep referring to paper [15] but I think they should be referring to paper [11] instead. Am I right? Or [15] is correct?

You are correct, thank you and reviewer #2 both for bringing this to our attention. We mistakenly cited [15] rather than [11] since they were by the same first author and same publication year. We have now corrected these citations.

> 8. Table 1 is odd. I would expect this table to be the first part of the results of current point 3.4. This table should be intended to provide descriptive statistics of the participants, indicating Mean and SD for continuous variables, and number of observations and percentages n(%) for categorical variables. There is no heading. Include race/ethnicity information, and other sociodemographic variables if available. We need to know what type of participants the manuscript is analyzing. Include diagnoses too.

Thank you for these suggestions! We have collected ethnicity and diagnostic information for our participants, and these are now displayed in Table 1a on pages 9-10 (previously Table 1), which is now organized by headings such as "Ethnicity", "Diagnosis", etc.

Ideally, the table should include descriptive statistics for all variables employed in the models.

We agree with the reviewer’s suggestion that this added information is valuable. Hence, we have added a new table, Table 1c on pages 11-12, that contains these statistics.

> The most important are temper outbursts, but the information provided for these is unclear (Mean and SD of frequency?). What is the denominator of such frequency? I would be trans

---

## [Decision Letter · Decision Letter 2]

28 Dec 2024

PONE-D-23-20323R2Multivariate prediction of temper outbursts in an observational study of youth enriched for irritability using Ecological Momentary Assessment dataPLOS ONE

Dear Dr. Zheng,

Thank you for submitting your manuscript to PLOS ONE. After careful consideration, we feel that it has merit but does not fully meet PLOS ONE’s publication criteria as it currently stands. Therefore, we invite you to submit a revised version of the manuscript that addresses the points raised during the review process.

**ACADEMIC EDITOR: ** Please address the queries of reviewer 3 in your response letter, and make necessary edits in your revised manuscript.With regards to reviewer 2, a senior PLoS One editor has confirmed that the report is permitted as per PLoS guidelines.

We look forward to receiving your revised manuscript.

Kind regards,

Sandip Varkey George, PhD

Academic Editor

PLOS ONE

Journal Requirements:

Reviewers' comments:

Reviewer's Responses to Questions

**Comments to the Author**

1. Does the manuscript provide a valid rationale for the proposed study, with clearly identified and justified research questions?

Reviewer #2: Partly

Reviewer #3: Yes

2. Is the protocol technically sound and planned in a manner that will lead to a meaningful outcome and allow testing the stated hypotheses?

Reviewer #2: Partly

Reviewer #3: Yes

3. Is the methodology feasible and described in sufficient detail to allow the work to be replicable?

Reviewer #2: No

Reviewer #3: Yes

4. Have the authors described where all data underlying the findings will be made available when the study is complete?

Reviewer #2: Yes

Reviewer #3: Yes

5. Is the manuscript presented in an intelligible fashion and written in standard English?

Reviewer #2: No

Reviewer #3: Yes

6. Review Comments to the Author

You may also provide optional suggestions and comments to authors that they might find helpful in planning their study.

Reviewer #2: With the new edits, I can see this protocol has already collected its data, and therefore is not a registered report protocol.

Reviewer #3: The authors have addressed most of my comments. Most importantly, with the new structure, the manuscript is much easier to understand and follow. Yet, after re-reading the authors' response to referees' comments and the revised manuscript, I still have some minor suggestions that I think will make this work much clearer.

1.Re: Title. Perhaps it should be specified, instead of observational study, that this is a registered report. It is named like this by the authors in the background, point 1.1., and discussion. In addition, youth can not be enriched for irritability; samples of youth are. My suggestion for the title is therefore, “Multivariate prediction of temper outbursts in a sample of youth enriched for irritability using Ecological Momentary Assessment data: a registered report”

2.As new references have been included (i.e. 30 to 39) these are no longer in order. References 30 to 38 (skipping 32) are cited between references 13 and 14. References 32 and 33 are listed between references 7 and 8. References 30 and 31 are cited between references 37 and 38. Refence 39 is also in the wrong order. Authors should update the order of references.

3.It is unclear what the section “1.1. Organization of this Registered report” is. This section is blank, jumping straight to the next section “1.2. Related work” I believe there must be a confusion on the hierarchy of sections that should be corrected.

4.Just under current section “2.2. Preliminary data”, when describing the design of the study, I believe authors should list 1197 prompts (which is what is intended) rather than 932 prompts, which is what they eventually analyzed. These 932 prompts are rightfully explained later in the report.

5.I suggest the authors to name Tables 1a, 1b, and 1c, as Tables 1, 2 and 3. And then Tables 2 and 3, as Tables 4, and 5 respectively.

6.In current Table 1a, authors should provide SD for mean age.

7.“Grouchy” is described as an item that collected irritability-related impairment. Isn’t this an item that, along with irritable mood, captures tonic irritability rather than impairment? Indeed, according to the protocol in Naim et al 2021, the irritability-related impairment item is described as follows: [“Since the last beep, my grouchy mood, or being angry and out of control, got me in trouble” (categorical “with my parent,” “at school,” and/or “with other kids” [multiple selections allowed], or “none of the above”). The total number of domains of impairment was calculated for each participant (range = 0–3).] Authors should make sure that this item is accurately described.

8.Under section “2.3. Methods for classifier development on preliminary data” “Variables used for prediction” authors mention that they explored two different lag setting. They describe lag-1 but I missed the description of lag-2. Can they describe it?

9.Related to previous point, later in the manuscript under section “2.4.2. Logistic regression” authors mention that they determined the optimal number of time lags, which includes 1, 2 or 3. However, when declaring the analyses they only mention that explored 1 and 2, not 3. The authors should declare and describe lag 3 as well.

10.Related, the authors say that selected lag-2, but then, if I understood well the output, they present the results of lag 1 and lag 2 in Table 3, not just lag 2, and the confirmatory analysis also includes lag 1, even though lag 3 has a higher AUC than lag 1. Authors should clarify this point.

11.Can the authors describe/interpret the results from current Table 2 and 3 for the reader? What do they tell us about the prediction of temper outbursts, or lack of it, and the predictive value of each variable?

12.Under section “3.3.2. Analytic approach”, authors mention that “The variables coincident with a temper outburst are not used in the prediction because, given the design of the EMA application, temper outbursts will typically only be recorded after they have already occurred.” This should be mentioned under section 2.3, subsection “Variables used for prediction”

7. PLOS authors have the option to publish the peer review history of their article (what does this mean? ). If published, this will include your full peer review and any attached files.

**Do you want your identity to be public for this peer review?** For information about this choice, including consent withdrawal, please see our Privacy Policy .

Reviewer #2: No

Reviewer #3: **Yes: ** Pablo Vidal-Ribas

---

## [Author Response · Author response to Decision Letter 3]

28 Jan 2025

Jan 27, 2025

Response letter

To the academic editor and reviewers:

We are thankful to the academic editor and the reviewer for their insightful comments.

Response to academic editor

> Please address the queries of reviewer 3 in your response letter, and make necessary edits in your revised manuscript.

With regards to reviewer 2, a senior PLoS One editor has confirmed that the report is permitted as per PLoS guidelines.

Indeed, we have edited the manuscript to reflect many of the excellent suggestions by reviewer 3. We greatly appreciate the support of PLoS One in confirming the

suitability of our manuscript for the registered report format.

Response to reviewer #2

> With the new edits, I can see this protocol has already collected its data, and therefore is not a registered report protocol.

This is a very reasonable objection, as it was also made by the PLoS ONE editors when the manuscript was initially submitted. Since this discussion occurred prior to our manuscript's first review, we appreciate that reviewer #2 did not have access to this discussion. Therefore, we are including the details below.

Our initial submission was rejected by the editor on the following grounds:

> "Registered Report Protocols submitted to PLOS ONE should report the study proposal prior to conducting experiments and/or data collection (https://journals.plos.org/plosone/s/other-article-types#loc-registered-reports). In this case, we note that experiments and/or observations have already begun. Preliminary or pilot data may be included, but only if necessary to support the feasibility of the study or as a proof of principle. As such, this submission does not meet our criteria for registered report protocols."

We appealed the decision and defended the use of preliminary data in the following response.

>"...we believe there has been a misunderstanding. Specifically, we do not consider the preliminary data that we report to be part of the proposed study—rather, the proposed study will begin with the data we propose to collect for the confirmatory analysis, which will happen in the future.

The preliminary data was collected for another study, and we used it primarily to develop a predictive model. In our experience, it is most productive to first develop a predictive model before pre-registration, because model development often reveals unknown features of the data. Yet, to ensure that the model performance can be evaluated on an independent dataset, it is beneficial to pre-register the collection and analysis of a confirmatory data set. We realize that this is a subtle distinction, and if it is acceptable to the journal, we would like to resubmit after clarifying the difference between the preliminary data and the data to be collected for the protocol."

In response, the editorial team rescinded the rejection, and gave us the opportunity to re-submit under the following condition:

> "Please update the submission to clarify the preliminary nature of the data presented and the difference between that preliminary data and the dataset subject to the preregistration as described in your appeal request. "

Therefore, we updated the submission to clarify the difference between the preliminary dataset, which we already possess, and the confirmatory dataset that we plan to collect. The journal then initiated the review process for the manuscript.

We hope that these details will be helpful in clarifying why our manuscript is suitable for the registered report format, despite the large amount of already collected data.

Response to reviewer #3

The authors have addressed most of my comments. Most importantly, with the new structure, the manuscript is much easier to understand and follow. Yet, after re-reading the authors' response to referees' comments and the revised manuscript, I still have some minor suggestions that I think will make this work much clearer.

> 1.Re: Title. Perhaps it should be specified, instead of observational study, that this is a registered report. It is named like this by the authors in the background, point 1.1., and discussion. In addition, youth can not be enriched for irritability; samples of youth are. My suggestion for the title is therefore, “Multivariate prediction of temper outbursts in a sample of youth enriched for irritability using Ecological Momentary Assessment data: a registered report”

Thank you for this suggestion. We have changed the title accordingly.

> 2.As new references have been included (i.e. 30 to 39) these are no longer in order. References 30 to 38 (skipping 32) are cited between references 13 and 14. References 32 and 33 are listed between references 7 and 8. References 30 and 31 are cited between references 37 and 38. Refence 39 is also in the wrong order. Authors should update the order of references.

Thank you for your feedback. We have renumbered the references.

> 3.It is unclear what the section “1.1. Organization of this Registered report” is. This section is blank, jumping straight to the next section “1.2. Related work” I believe there must be a confusion on the hierarchy of sections that should be corrected.

Our apologies for the confusion. We inadvertently omitted the text for Section 1.1, which describes the organization of the paper into 4 sections.

> 4.Just under current section “2.2. Preliminary data”, when describing the design of the study, I believe authors should list 1197 prompts (which is what is intended) rather than 932 prompts, which is what they eventually analyzed. These 932 prompts are rightfully explained later in the report.

Thank you for the suggestion. We have changed the sentence accordingly.

> "We analyzed a preliminary set of data collected starting July 2018 and ending in December 2022, in which n=57 participants (with a diagnosis of either ODD, DMDD, or sub-threshold DMDD) completed EMA, with a total of 1197 prompts (3 prompts per day for 7 days, up to 21 completed prompts per participant depending on completion rate)."

> 5.I suggest the authors to name Tables 1a, 1b, and 1c, as Tables 1, 2 and 3. And then Tables 2 and 3, as Tables 4, and 5 respectively.

Thank you for the suggestion. We have renumbered the tables.

> 6.In current Table 1a, authors should provide SD for mean age.

Thank you for noticing our omission. We have added the SD for mean age.

> 7.“Grouchy” is described as an item that collected irritability-related impairment. Isn’t this an item that, along with irritable mood, captures tonic irritability rather than impairment? Indeed, according to the protocol in Naim et al 2021, the irritability-related impairment item is described as follows: [“Since the last beep, my grouchy mood, or being angry and out of control, got me in trouble” (categorical “with my parent,” “at school,” and/or “with other kids” [multiple selections allowed], or “none of the above”). The total number of domains of impairment was calculated for each participant (range = 0–3).] Authors should make sure that this item is accurately described.

Thank you for bringing this to our attention. We mistakenly listed the same "grouchy" item twice and confused it with the impairment-related item in Naim et al. 2021, which was not included in our study. We have deleted the redundant mention of this item which incorrectly associates this item with impairment.

> 8.Under section “2.3. Methods for classifier development on preliminary data” “Variables used for prediction” authors mention that they explored two different lag setting. They describe lag-1 but I missed the description of lag-2. Can they describe it?

Thanks for noticing -- we indeed intended to describe lag-2 but omitted it. We have now added the description of lag-2 and lag-3 on page 16:

> "The "lag-2" model incorporates all the variables used by the "lag-1" model and adds to it all the variables from the prompt preceding the previous prompt. For example, to predict the temper outburst response on day 3 morning, we use the responses from day 2 evening and day 2 afternoon. Meanwhile, the "lag-3" model adds the variables from three prompts ago in addition to all the variables from the "lag-2" model. Continuing with our example, to predict the temper outburst response on day 3 morning, we use the responses from day 2 morning, in addition to the responses from day 2 afternoon (also used by "lag-2") and day 2 evening (also used by "lag-2" and "lag-1")."

> 9.Related to previous point, later in the manuscript under section “2.4.2. Logistic regression” authors mention that they determined the optimal number of time lags, which includes 1, 2 or 3. However, when declaring the analyses they only mention that explored 1 and 2, not 3. The authors should declare and describe lag 3 as well

Thank you for this suggestion. As we mentioned in our response to the previous item, we have now added a description of "lag-2" and "lag-3" on page 16. Furthermore, we have now included model coefficients for all three models in the supplement S2 appendix.

> 10.Related, the authors say that selected lag-2, but then, if I understood well the output, they present the results of lag 1 and lag 2 in Table 3, not just lag 2, and the confirmatory analysis also includes lag 1, even though lag 3 has a higher AUC than lag 1. Authors should clarify this point.

Thank you for noticing this need for clarification. We are not using the lag-1 model in the confirmatory analysis, but we see that our writing could give that impression, since we said that we include "lags 1 and 2". This is referring to the fact that the "lag-2" model includes both lags 1 and 2 in a single model. We have clarified this in our revision on page 26:

> "We will use the "lag-2" model as described in section 2.3. Hence, time-varying predictors will be included with both lags of 1 and 2, so that information in the two measurement periods proceeding a temper outburst is used to predict the absence/presence of a temper outburst."

> 11.Can the authors describe/interpret the results from current Table 2 and 3 for the reader? What do they tell us about the prediction of temper outbursts, or lack of it, and the predictive value of each variable?

We have refrained from over-interpreting the preliminary results, since the current sample size is not enough for many of these parameters to be credibly nonzero (Table 2). Furthermore, since the hypothesis we wish to test concerns the feasibility of the predictive model, we would like to avoid confusing the reader of the purpose of the pre-registration by prematurely speculating on the variables in the model. We are extremely interested in the interpretation of the model weights and would be more comfortable doing so in the post-hoc section of the confirmatory study.

> 12.Under section “3.3.2. Analytic approach”, authors mention that “The variables coincident with a temper outburst are not used in the prediction because, given the design of the EMA application, temper outbursts will typically only be recorded after they have already occurred.” This should be mentioned under section 2.3, subsection “Variables used for prediction”

Thank you for the suggestion. We have moved the sentences to section 2.3.

---

## [Editor Report · Decision Letter 3]

7 Feb 2025

Multivariate prediction of temper outbursts in a sample of youth enriched for irritability using Ecological Momentary Assessment data: a registered report

PONE-D-23-20323R3

Dear Dr. Zheng,

We’re pleased to inform you that your manuscript has been judged scientifically suitable for publication and will be formally accepted for publication once it meets all outstanding technical requirements.

Kind regards,

Sandip V George, PhD

Academic Editor

PLOS ONE

Additional Editor Comments (optional):

Thanks for your submission, and in my view the revised manuscript can be accepted as a registered report. All the best with the upcoming analysis.
---

## [Editor Report · Acceptance letter]

PONE-D-23-20323R3

PLOS ONE

Dear Dr. Zheng,

I'm pleased to inform you that your manuscript has been deemed suitable for publication in PLOS ONE. Congratulations! Your manuscript is now being handed over to our production team.

Kind regards,

on behalf of

Dr. Sandip V George

Academic Editor

PLOS ONE